# Robust skeletal motion tracking using temporal and spatial synchronization of two video streams

Vytautas Abromavičius[1,2]*, Ervinas Gisleris[2], Kristina Daunoravičienė[3], Jurgita Žižienė[3], Artūras Serackis[2], Rytis Maskeliūnas[1]

**1** Faculty of Informatics, Kaunas University of Technology, Kaunas, Lithuania, **2** Department of Electronic Systems, Vilnius Gediminas Technical University, Vilnius, Lithuania, **3** Department of Biomechanical Engineering, Vilnius Gediminas Technical University, Vilnius, Lithuania

* vytautas.abromavicius@ktu.lt

**Data availability statement:** Data cannot be shared publicly because of confidentiality

## Abstract

Accurate and reliable skeletal motion tracking is essential for rehabilitation monitoring, enabling objective assessment of patient progress and facilitating telerehabilitation applications. Traditional marker-based motion capture systems, while highly accurate, are costly and impractical for home rehabilitation, whereas marker-less methods often suffer from depth estimation errors and occlusions. Recent studies have explored various computer vision and deep learning approaches for human pose estimation, yet challenges remain in ensuring robust depth accuracy and tracking under occlusion conditions. This study proposes a three-dimensional human skeleton tracking system for upper limb activities that integrates temporal and spatial synchronization to improve depth estimation accuracy for rehabilitation exercises. The proposed system combines a 90° secondary camera to compensate for the depth prediction inaccuracies inherent in single-camera systems, reducing error margins by up to 0.4 m. In addition, a linear regression-based depth error correction model is implemented to refine depth coordinates, further improving tracking precision. The Kalman filtering framework is employed to enhance temporal consistency, allowing real-time interpolation of missing joint positions. Experimental results demonstrate that the proposed method significantly reduces depth estimation errors of the elbow and wrist joint ($p < 0.001$) compared to single camera setups, particularly in scenarios involving occlusions and non-frontal perspectives. This study provides a cost-effective and scalable solution for remote patient monitoring and motor function evaluation.

## Introduction

Human pose estimation is emerging in rehabilitation patient monitoring, using advances in artificial intelligence and computer vision to improve patient care. The tracking of human movement kinematics offers significant potential to improve motor assessment and telerehabilitation for stroke patients [1]. Various methodologies have been explored, including using YOLOv4-tiny for patient detection and Mediapipe for pose estimation, achieving high

obligations governing human subject research mandated by the Ethics Committee (provided in Ethics statement). Joint estimations and corrected errors are provided and publicly available from https://doi.org/10.5281/zenodo.15075245. Other data are available for researchers who meet the criteria for access to confidential data. Name of the Restricting institution is Vilnius Gediminas Technical University (VILNIUS TECH). Point of contact to access data is Julius Griškevičius, responsible officer for project data governance, Vilnius Gediminas Technical University.

**Funding:** This project has received funding from the Research Council of Lithuania (LMTLT), agreement No S-PD-24-29. The funders had no role in study design, data collection and analysis, decision to publish, or preparation of the manuscript.

**Competing interests:** The authors have declared that no competing interests exist.

accuracy in classifying patient poses such as sleeping, sitting, walking and standing [2]. Despite the challenges posed by complex environmental changes and diverse body shapes, advanced techniques such as coarse-to-fine heatmap shrinking and spatial-temporal perception networks have shown promise in enhancing the accuracy of 2D and 3D pose estimations in clinical settings [3,4]. Virtual rehabilitation systems, such as those based on PoseNet, enable patients to perform rehabilitation exercises at home, with real-time tracking of joint movements to assess recovery progress [5]. The integration of low-cost and high-end sensor data, as demonstrated by datasets containing extensive motion sequences, further supports the development of robust inertial pose estimation algorithms for rehabilitation applications [6]. Furthermore, the fusion of binocular stereo vision and convolutional neural networks has been explored to improve the accuracy of 3D pose estimation, potentially reducing costs and limitations associated with traditional motion capture systems [7], also capable of converting limited angle sensor data to a full 3D image of the pose tracked [8]. Smart walkers with RGB+D cameras and neural network frameworks offer a comprehensive approach to estimating full-body poses, facilitating real-time monitoring and human-robot interaction in rehabilitation settings [9]. The combination of inertial measurement units with computer vision techniques has been identified as a promising direction for future research, with the aim of improving the effectiveness of rehabilitation therapy [10]. In addition, Virtual Reality approaches and sensors-based techniques can lead to more precise prose tracking, further improving the precision of therapy [11].

Over the past decade, deep learning has significantly advanced in detecting and tracking human body parts from images or videos to build a representation of the human body, often based on convolutional neural networks (CNNs) and other architectures to improve accuracy and efficiency. Various datasets, such as the MPII Human Pose dataset, have been collected to train and evaluate these models, with notable implementations, including ResNet50 and VGG16, which achieved 67% and 88.8% accuracy, respectively [12]. Despite these advances, challenges such as insufficient training data, depth ambiguities, and occlusions persist, which requires ongoing research and development [13]. Applications of pose estimation span numerous domains, including human-computer interaction, motion analysis, augmented reality, and virtual reality [14]. Methods for estimating poses typically involve bottom-up or top-down approaches, CNN being a common choice for detecting key points, while movement assessment methods vary widely [15]. Approaches, such as the Ultra Wide-Band (UWB) technology with body-mounted sensors, have also been explored to overcome the limitations of vision-based systems, particularly in complex environments [16]. Furthermore, models such as BlazePose GHUM 3D and HRNet have been developed to enhance the accuracy of 2D and 3D pose estimation, addressing problems related to image resolution and multiscale problems [3,17]. Despite progress, achieving precision of 100% remains elusive due to factors such as environmental changes and various body shapes [18]. The field continues to evolve with ongoing research that focuses on improving model robustness, dataset diversity, and real-time application capabilities [19,20]. Future directions include refining feedback mechanisms and addressing erroneous feedback to enhance user interaction and application effectiveness.

Movement analysis allows for the early detection of deviations from standard movement patterns and provides accurate and objective data on correct limb movements, posture, body balance, and coordination. Objective qualitative and quantitative insights allow the correct treatment plans to be tailored to the specific needs of each subject for objectified movement assessment [21]. However, those that exist are time-consuming and unfriendly to the researcher and the subject. Table 1 summarizes recent research models in human pose estimation on a Human3.6M dataset.

**Table 1. Summary of models for 3D human pose estimation with datasets and performance metrics.**

| Model | Year | Key Features | Datasets | Performance Metrics |
|---|---|---|---|---|
| **PoseFormer** [23] | 2021 | Transformer-based approach for 3D human pose estimation, utilizes a spatial-temporal attention mechanism. | Human3.6M, MPI-INF-3DHP | MPJPE: 51.4 mm on Human3.6M |
| **HybrIK** [24] | 2021 | Hybrid model integrating Inverse Kinematics with a neural network, improves joint accuracy and articulation. | Human3.6M, COCO | MPJPE: 41.6 mm on Human3.6M |
| **Mesh Graphormer** [25] | 2022 | Graph convolutional network combined with transformer, focuses on human mesh recovery and pose estimation. | Human3.6M, 3DPW | PA-MPJPE: 45.6 mm on 3DPW |
| **Pose2Mesh** [26] | 2021 | End-to-end framework converting 2D pose to 3D mesh, employs pose refinement with mesh regression. | Human3.6M, 3DPW | PA-MPJPE: 47.9 mm on 3DPW |
| **LiftFormer** [27] | 2022 | Lifting-based method using transformers for lifting 2D poses to 3D, emphasizes temporal consistency. | Human3.6M, MPI-INF-3DHP | MPJPE: 40.2 mm on Human3.6M |
| **GAST-Net** [28] | 2022 | Geometry-aware spatiotemporal network, integrates geometric priors with spatiotemporal reasoning. | Human3.6M, 3DPW | MPJPE: 39.8 mm on Human3.6M |
| **Sosa-self** [29] | 2023 | Three network ensemble. | Human3.6M, 3DPW | MPJPE: 43.4 mm on Human3.6M |
| **SlowFastFormer** [30] | 2024 | Network where two branches with different input rates are composed to encode these two different kinds of context. | Human3.6M | MPJPE: 38.9 mm on Human3.6M |
| **Hourglass Tokenizer** [31] | 2024 | Trims and then restores tokens in a transformer. | Human3.6M, MPI-INF-3DHP | MPJPE: 39 mm on Human3.6M |
| **MotionAGFormer** [32] | 2024 | Combines parallel transformer and CNN based streams to fuse global and local pose cues. | Human3.6M | MPJPE: 38.4 mm on Human3.6M |
| **This paper (adapted to single "camera")** | 2025 | Kalman filtering | Human3.6M | MPJPE: 41.2 mm on Human3.6M |

## Goal of the work

The research presented in this work is a continuation of our previous study on enhanced human skeleton tracking for rehabilitation exercises [22]. In the previous work, we used a dual-camera setup to mitigate depth prediction errors by combining two synchronized video streams. In this extended study, we further refine the system by integrating Kalman filtering for depth error correction and spatio-temporal synchronization, which enhances pose estimation accuracy across diverse movement patterns. Furthermore, we expanded the evaluation to include benchmark comparisons with state-of-the-art 3D pose estimation models on Human3.6M and private datasets.

The goal of this work is to develop a timeline and a spatially synchronized two-camera system to detect and predict the depth of specific joints, when volunteers perform various exercises for rehabilitation purposes [33] and to evaluate the accuracy of depth coordinate predictions, with concurrent movement tracking achieved through a specialized optical system. In addition, our objective was to identify exercises during which specific joints were difficult to detect and exercises in which the corresponding joints exhibited the highest depth prediction errors.

Building on our single-camera baseline [8], we attach an orthogonal side camera to address the dominant source of error, depth ambiguity. The two RGB streams are brought into the spatiotemporal register with an analytic three-step synchronization; a lightweight linear regression corrector then removes the residual per joint depth bias. Recent multiview works [34–36] focus on heavier networks for pose refinement but leave depth bias largely

untreated, whereas our explicit corrective model targets that error directly. Robustness is further strengthened by a Kalman smoother that bridges occlusions when either camera momentarily loses sight of a limb. Furthermore, we repurpose our earlier Pareto optimization scheme to select an operating point on the latency-versus-alignment front, ensuring real-time performance without retraining. In conclusion, we provide a detailed evaluation dedicated to upper limb rehabilitation, where seven shoulder-elbow-wrist movements are compared under occlusion and clothing variation.

## Materials and methods

### Implementation

Depth error is the biggest problem in predicting the 3D human skeleton from a single image stream. We propose including an additional video stream perpendicular to the first video stream. When filming a person from the side, an additional video stream will provide additional depth information, with the help of which it will be possible to adjust the coordinates of the predicted three-dimensional human skeleton obtained from the frontal video stream; however, it becomes mandatory to synchronize the video streams. Ideally, both video streams should be shot with the same cameras, ensuring the same frame quality, image resolution, and frame rate.

The diagram of the proposed solution algorithm is presented in Fig 1. The algorithm starts with two parallel video streams: one capturing a person from the front and the other from the side. The coordinates of the second skeleton are rotated 90° around the Y-axis to align the viewing positions. Depth error dependence between the skeletons is calculated and missing joint points are inserted from one stream to another, with depth adjustments based on the calculated errors. The final 3D skeleton is formed using the X and Y coordinates from the first stream and the Z coordinates from the second.

Video Stream Recording is defined as follows:

- Let $V_f(t)$ represent the video stream recording a person from the front over time $t$.
- Let $V_s(t)$ represent the video stream recording a person from the side over time $t$.

In the 3D human skeleton prediction stage of the proposed algorithm, the MediaPipe Pose system, improved on the work of [37], was selected as an open source backbone due to its real-time processing capability, and was further modified to provide sufficient accuracy and the provision of a higher number of 3D human skeleton coordinates compared to previous solutions [34].

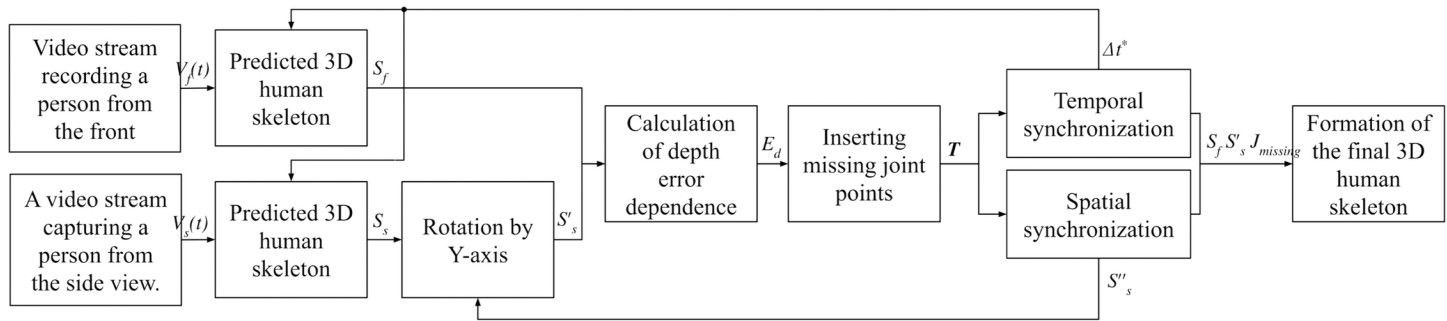

**Fig 1. Diagram of the proposed solution algorithm.**

The prediction of depth coordinates from a single image often exhibits errors that correlate with the vertical position of the joints. To address this, we implemented a linear regression model to predict and correct for these errors. Within the "Depth error dependence calculation block", we determined the coefficients k and b for the linear relationship:

$$y = k \cdot z + b, \tag{1}$$

where $y$ represents the vertical coordinate of the human joints and $z$ is the predicted depth coordinate. The calculation of the depth prediction error at a given height $y$ is defined as follows:

$$z = \frac{y - b}{k}. \tag{2}$$

The algorithm aims to minimize prediction error by optimizing coefficients $k$ and $b$, thus enhancing the overall accuracy and efficiency of the 3D prediction of the human skeleton. Our method ensures that the linear regression model accurately reflects the relationship between the vertical and depth coordinates, facilitating more precise depth predictions. We used this depth coordinate prediction for only visible joints from one camera.

Using the above principle behind the pose estimation algorithms, we predict the 3D human skeleton:

$$S_f = \text{PoseEstimation}(V_f(t)), \tag{3}$$

$$S_s = \text{PoseEstimation}(V_s(t)). \tag{4}$$

We perform a rotation on the Y axis to align the side view skeleton $S_s$ with the front view skeleton $S_f$.

$$S'_s = \text{RotateY}(S_s, \theta), \tag{5}$$

where $\theta$ is the rotation angle.

Depth error $E_d$ is calculated as the difference between the depths of the corresponding joints in $S_f$ and $S'_s$.

$$E_d = \sum_{i=1}^{n} \left| J_f^z(i) - J_s'^z(i) \right|, \tag{6}$$

where $n$ is the number of joints and $z$ denotes the depth coordinate. $J_f(i)$ and $J'_s(i)$ represent the joint points in the skeletons $S_f$ and $S'_s$ respectively. If a joint point $J_f(i)$ is missing, it is inserted from $J'_s(i)$ and vice versa. The insertion process is enhanced using Kalman filtering and depth error correction.

We adapt the state vector for the Kalman filter as suggested by [38] as:

$$\mathbf{x}_k = \begin{bmatrix} x_k \\ y_k \\ z_k \\ \dot{x}_k \\ \dot{y}_k \\ \dot{z}_k \end{bmatrix}, \tag{7}$$

where $x_k, y_k, z_k$ are the coordinates of the joint point and $\dot{x}_k, \dot{y}_k, \dot{z}_k$ are the velocities. The Kalman filter is applied under the assumption that joint trajectories exhibit approximately

linear motion over each frame, making a first-order linear model suitable for state prediction. Furthermore, previous studies on pose estimation suggest that measurement noise follows a Gaussian distribution [39].

The prediction step is then defined as:

$$\mathbf{x}_{k|k-1} = \mathbf{A}\mathbf{x}_{k-1|k-1} + \mathbf{B}\mathbf{u}_k, \tag{8}$$

where $\mathbf{A}$ is the state transition matrix and $\mathbf{B}$ is the control input matrix. We assume the zero control input ($\mathbf{B} = 0$, $\mathbf{u}_k = 0$) and a constant velocity for the state transition matrix $\mathbf{A}$, where the position of each joint is updated as a function of its previous position and estimated velocity:

$$A = \begin{bmatrix} 1 & 0 & 0 & \Delta t & 0 & 0 \\ 0 & 1 & 0 & 0 & \Delta t & 0 \\ 0 & 0 & 1 & 0 & 0 & \Delta t \\ 0 & 0 & 0 & 1 & 0 & 0 \\ 0 & 0 & 0 & 0 & 1 & 0 \\ 0 & 0 & 0 & 0 & 0 & 1 \end{bmatrix}. \tag{9}$$

The update step is defined as:

$$\mathbf{K}_k = \mathbf{P}_{k|k-1}\mathbf{H}^T(\mathbf{H}\mathbf{P}_{k|k-1}\mathbf{H}^T + \mathbf{R})^{-1}, \tag{10}$$

$$\mathbf{x}_{k|k} = \mathbf{x}_{k|k-1} + \mathbf{K}_k(\mathbf{z}_k - \mathbf{H}\mathbf{x}_{k|k-1}), \tag{11}$$

$$\mathbf{P}_{k|k} = (\mathbf{I} - \mathbf{K}_k\mathbf{H})\mathbf{P}_{k|k-1}, \tag{12}$$

where $\mathbf{K}_k$ is the Kalman gain, $\mathbf{P}_{k|k-1}$ is the predicted error covariance, $\mathbf{H}$ is the observation matrix, and $\mathbf{R}$ is the observation noise covariance. The observation model $\mathbf{H}$ is formulated as a linear mapping between the internal state of the system and the measured joint positions. In this case, an identity transformation is used. This simplifies the measurement update step. $\mathbf{R}$ was obtained by recoding a static subject for 30 s (no motion), and computing sampled variances of the detected joints:

$$\mathbf{R} = \text{diag}(\sigma_x^2, \sigma_y^2, \sigma_z^2). \tag{13}$$

In our filter, this fully specifies the measurement update weighting. A small process-noise term $\mathbf{Q} = 10^{-6}\,\mathbf{I}$ was included for numerical stability.

The filter was initialized with

$$\mathbf{x}_{0|0} = \begin{bmatrix} x_0 & y_0 & z_0 & 0 & 0 & 0 \end{bmatrix}^T, \qquad \mathbf{P}_{0|0} = \mathbf{I}. \tag{14}$$

The sampling interval was set to

$$\Delta t = \frac{1}{30}\ \text{s}, \tag{15}$$

matching the 30 Hz frame rate.

When a missing point is inserted, the depth value $z$ is adjusted based on depth error $E_d$. We then define the adjusted depth $z_{adjusted}$ as:

$$z_{adjusted} = z + \alpha E_d, \tag{16}$$

where $\alpha$ is a scaling factor.

Using the Kalman filter and depth error correction, the missing joint point $J_{missing}$ is interpolated as:

$$J_{missing} = \text{KalmanFilter}\left(\text{Interpolate}(S_f, S_s')\right) + \alpha E_d. \tag{17}$$

The final formation of the Final Three-Dimensional Human Skeleton is done by combining $S_f$, $S_s'$, and $J_{missing}$ to form the final 3D skeleton $S_{final}$:

$$S_{final} = \text{Combine}(S_f, S_s', J_{missing}). \tag{18}$$

The two camera streams $V_f(t)$ and $V_s(t)$ for the skeletons $S_f$ and $S_s$ must be *spatially synchronized*. To align the coordinate systems of the front view $S_f$ and the side view $S_s$, a translation vector $\mathbf{T}$ is introduced. This vector compensates for the different spatial positions of the cameras:

$$S_s'' = \text{Translate}(S_s, \mathbf{T}), \tag{19}$$

where $S_s''$ is the spatially adjusted side view skeleton and $\mathbf{T}$ is the translation vector defined as:

$$\mathbf{T} = \begin{bmatrix} T_x \\ T_y \\ T_z \end{bmatrix}, \tag{20}$$

where $T_x$, $T_y$, and $T_z$ are the translation offsets in the $x$, $y$, and $z$ axes, respectively.

Following translation, the side view skeleton $S_s''$ is rotated to align with the front view skeleton $S_f$, similar to the previous rotation step:

$$S_s''' = \text{RotateY}(S_s'', \theta), \tag{21}$$

where $S_s'''$ is the skeleton after both translation and rotation adjustments.

After applying translation and rotation, a spatial synchronization error $E_{spatial}$ is calculated to quantify the misalignment between the adjusted side view skeleton $S_s'''$ and the front view skeleton $S_f$:

$$E_{spatial} = \sum_{i=1}^{n} \left\| \mathbf{J}_f(i) - \mathbf{J}_s'''(i) \right\|_2, \tag{22}$$

where $\mathbf{J}_f(i)$ and $\mathbf{J}_s'''(i)$ represent the joint points in the skeletons $S_f$ and $S_s'''$, respectively, and $\|\cdot\|_2$ denotes the Euclidean distance.

To minimize the spatial synchronization error $E_{spatial}$, an optimization process is carried out to find the optimal translation vector $\mathbf{T}^*$ and rotation angle $\theta^*$, so the side view skeleton is spatially aligned as closely as possible with the front view skeleton.:

$$(\mathbf{T}^*, \theta^*) = \arg\min_{\mathbf{T}, \theta} E_{spatial}. \tag{23}$$

With the optimal translation $\mathbf{T}^*$ and rotation $\theta^*$ applied, the side view skeleton $S_s'''$ is used instead of $S_s'$ in the existing pipeline. Finally, the steps involving depth error calculation, Kalman filtering, and joint point interpolation are then carried out using the spatially synchronized skeletons:

$$E_d = \sum_{i=1}^{n} \left| S_f^z(i) - S_s'''^z(i) \right|, \tag{24}$$

$$J_{missing} = \text{KalmanFilter}\left(\text{Interpolate}(S_f, S_s''')\right) + \alpha E_d. \tag{25}$$

The final 3D skeleton $S_{final}$ is formed by combining $S_f$, $S_s'''$, and $J_{missing}$:

$$S_{final} = \text{Combine}(S_f, S_s''', J_{missing}). \tag{26}$$

*A synchronization of the timeframe* of the two camera streams $V_f(t)$ and $V_s(t)$ is done using a Pareto optimization-based timeframe synchronization method, proposed to improve the accuracy of the final 3D skeleton $S_{final}$:

The timeframe adjustment variables are defined as $\Delta t_f$ and $\Delta t_s$ as timeframe adjustments for $V_f(t)$ and $V_s(t)$, respectively, representing small shifts in time to synchronize the frames between the two camera streams.

The objective function aims to minimize the synchronization error, which is the difference between the depth coordinates of the corresponding joints in the skeletons $S_f$ and $S_s'$ at the adjusted timeframes:

$$E_{sync}(\Delta t_f, \Delta t_s) = \sum_{i=1}^{n} \left| S_f^z(i, t + \Delta t_f) - S_s'^z(i, t + \Delta t_s) \right|. \tag{27}$$

Computational Cost Minimization is defined as:

$$C(\Delta t_f, \Delta t_s) = C_f(\Delta t_f) + C_s(\Delta t_s), \tag{28}$$

where, $C_f(\Delta t_f)$ and $C_s(\Delta t_s)$ represent the computational cost associated with adjusting the timeframes $\Delta t_f$ and $\Delta t_s$ for the front and side views, respectively.

Implemented Pareto optimization algorithm flowchat is shown in Fig 2. The Pareto optimization, based on [8] is applied to find the optimal set of timeframe adjustments $(\Delta t_f^*, \Delta t_s^*)$ that simultaneously minimize the synchronization error $E_{sync}$ and the computational cost $C$. The Pareto front represents the set of non-dominated solutions where improving one objective (reducing $E_{sync}$) results in the deterioration of the other objective (increasing $C$). From the Pareto front, the final selection of $\Delta t_f^*$ and $\Delta t_s^*$ is made based on the desired trade-off between the synchronization accuracy and the computational cost. Once the optimal timeframe adjustments $\Delta t_f^*$ and $\Delta t_s^*$ are determined, the adjusted streams $V_f(t + \Delta t_f^*)$ and $V_s(t + \Delta t_s^*)$ are used in the existing pose estimation and depth error correction pipeline:

$$\begin{aligned} S_f &= \text{PoseEstimation}(V_f(t + \Delta t_f^*)) \\ S_s &= \text{PoseEstimation}(V_s(t + \Delta t_s^*)). \end{aligned} \tag{29}$$

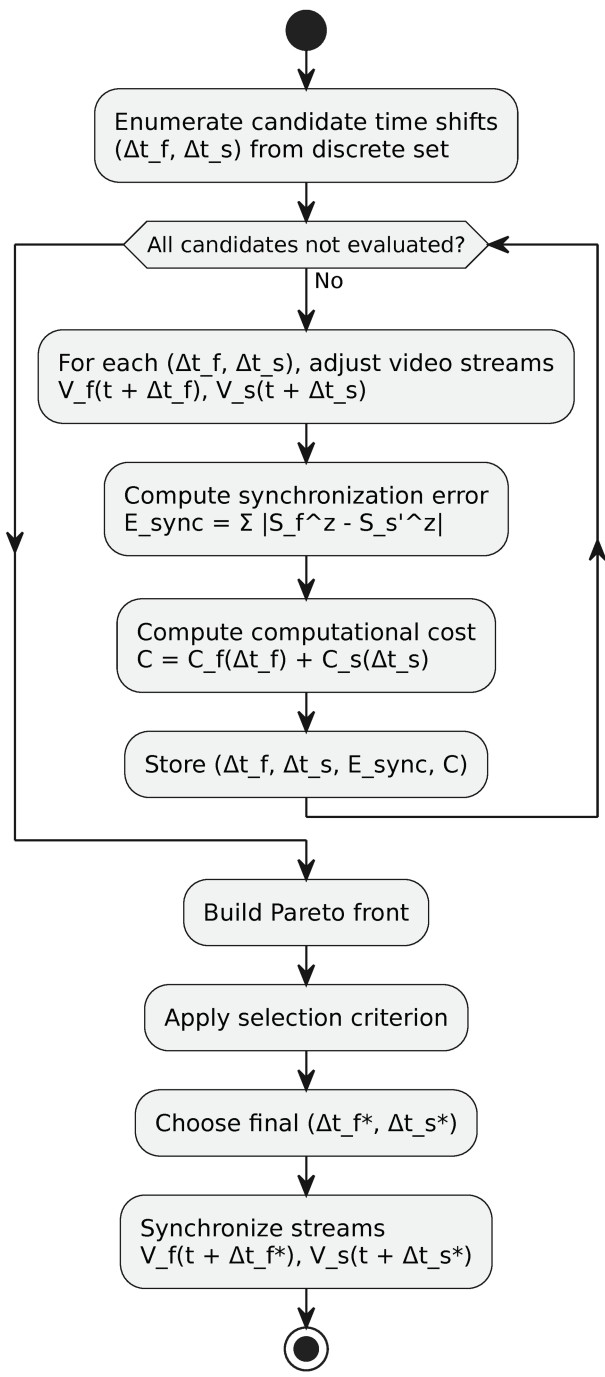

**Fig 2. Flowchart of the implemented Pareto optimization algorithm.**

The side view skeleton $S_s$ is rotated and aligned with the front view skeleton $S_f$, and the depth error $E_d$ is calculated as before. As described above, missing joint points are interpolated using Kalman filtering and depth error correction. The final 3D skeleton $S_{final}$ is formed by combining the synchronized skeletons $S_f$, $S_s'$, and $J_{missing}$.

## Dataset

Two datasets were collected in previous research by the authors [8]. Participants were recruited between 2023-06-09 and 2023-10-31. The study was carried out according to the Declaration of Helsinki and was approved by the Institutional Review Board of Vilnius Regional Biomedical Research Ethics Committee (no 2023/6-1439-978). Informed written consent was obtained from the participants. The datasets were collected using Gemini 2 cameras, which features a resolution of 1920×1080 at 30 frames per second for its RGB sensor. For processing and 3D human skeleton computation, the system relied on a high-performance PC equipped with an Intel i9-9900k processor, an NVIDIA RTX 4090 GPU with 24 GB of VRAM, and 64 GB of DDR4 RAM. The approach presented in this paper was developed using Python 3.11.2, and libraries NumPy 1.25.0, SciPy 1.11.1, and PyTorch 2.0.1.

Both cameras were mounted on tripods at approximately the same height ($\pm1$ cm) and oriented to form a nominal 90° angle. We approximate each camera as an ideal pinhole (principal point in the center of the image, no distortion coefficients) using the manufacturer's horizontal field of view to infer the focal length in pixels.

The first data set consists of nine pairs of videos of each person doing exercises dressed in black to have a baseline on camera limitations, grouped according to the exercise performed and the trial number, as presented in Table 2. Resolution of of the collected videos were $1920 \times 1080$.

Upper arm adduction while standing is performed from the initial position, when the volunteer is standing straight, hands down with palms facing the hips, and feet placed hip-width apart. The first exercise is performed by raising the right arm from the side until it is raised above the head. Then return to the starting position – the hand is lowered to the side, the palm facing the hips.

Standing flexion of the upper arm exercise is performed from the starting position, when the volunteer is standing straight, hands down with palms facing the hips, and feet placed hip-width apart. During the second exercise, the right hand is raised in front until it is raised above the head. The exercise is finished by lowering the arm and returning to the starting position.

Standing brachial adduction exercise is performed from the starting position, when the volunteer is standing straight, hands down with palms facing the hips, and feet placed hip-width apart. In the third exercise, the right hand is raised in front to the shoulder level and bent over the shoulder to the center of the face, the elbow always being extended. The exercise is completed by extending the arm back to shoulder level and lowering it to the side.

The second data set consists of eight pairs of videos per person. The recordings repeat the same exercises as in the first data set and additionally perform an exercise in which both arms are raised and lowered at the same time. The exercises are grouped according to the different outfits and the exercises performed and are presented in Table 3. Resolution of of the collected videos were $640 \times 480$.

**Table 2. Video summary of the first data set per person.**

| Exercise | Tests | Duration | Frames |
|---|---|---|---|
| Upper arm adduction while standing | 2 | 10 s | 345 |
| Standing flexion of the upper arm | 4 | 25 s | 786 |
| Standing brachial adduction | 3 | 15 s | 494 |

**Table 3. Video summary of the second data set per person.**

| Clothing | Exercise | Duration | Frames |
|---|---|---|---|
| Bright, Dark | Upper arm adduction while standing | 7, 6 s | 192, 206 |
| Bright, Dark | Standing flexion of the upper arm | 7, 7 s | 222, 219 |
| Bright, Dark | Standing brachial adduction | 6, 6 s | 168, 182 |
| Bright, Dark | Standing flexion of both arms | 7, 6 s | 220, 187 |

## Results

### Results using the 'First' dataset

The illustration in Fig 3 presents the predicted three-dimensional human skeleton during an arm flexion
exercise. The first two rows display five essential frames from the front and side video streams, respectively. Rows 3 and 4 present the reconstructed 3D skeleton: joints from the front camera appear as red filled circles (solid dark lines), whereas joints from the side camera are shown as hollow circles (light grey lines).

The illustration shows these results from three angles: side, front, and diagonal. Initially, the skeleton projected from the frontal video stream exhibits Z-axis estimation errors, evident as the upper body tilts forward and the lower body tilting backward. Additionally, the skeleton projected from the side stream in the second row lacks discernible joints on the left side of the human body. While our final proposed solution successfully straightens the 3D human skeleton and corrects right-hand movements, a depth error persists in the left hand.

Fig 4 illustrates the depth trajectory of the right elbow and wrist joints during an exercise of flexion of the upper arm. We selected these joints because they exhibit the most significant depth changes and are central to the exercise. The data highlight the impact of depth prediction errors in single-camera tracking and how our dual-camera approach with depth correction refines these estimates.

During the initial synchronization phase (frames 40–90), both video streams are aligned. From frame 100 onward, as the subject begins raising the arm, depth errors become more pronounced, especially at peak flexion. The raw depth estimation from the frontal camera shows a systematic bias, resulting in inconsistent joint positioning and a noticeable forward tilt of the upper body. However, after applying depth correction and Kalman filtering, the refined depth trajectory is smoother and more anatomically accurate, reducing fluctuations and aligning precisely with expected joint movements.

### Experimental results of the second part of the data set

The most significant predicted depth error occurs during the adduction and subsequent reextension of the forward-extended upper arm. Although the depth trajectories of the right elbow and wrist joints maintain similar patterns, their absolute depth values frequently diverge. Importantly, the refined joint depth trajectories are smoother and exhibit a reduced systematic bias, shifted closer to the center of the Y-axis.

The means and standard deviations of the predicted joint depth error for the first data set are shown in Table 4. The mean and standard deviation results are given in meters. The standard deviation was calculated as follows:

$$\sigma = \sqrt{\frac{1}{N}\left[(x_1 - \mu)^2 + (x_2 - \mu)^2 + \cdots + (x_N - \mu)^2\right]}, \tag{30}$$

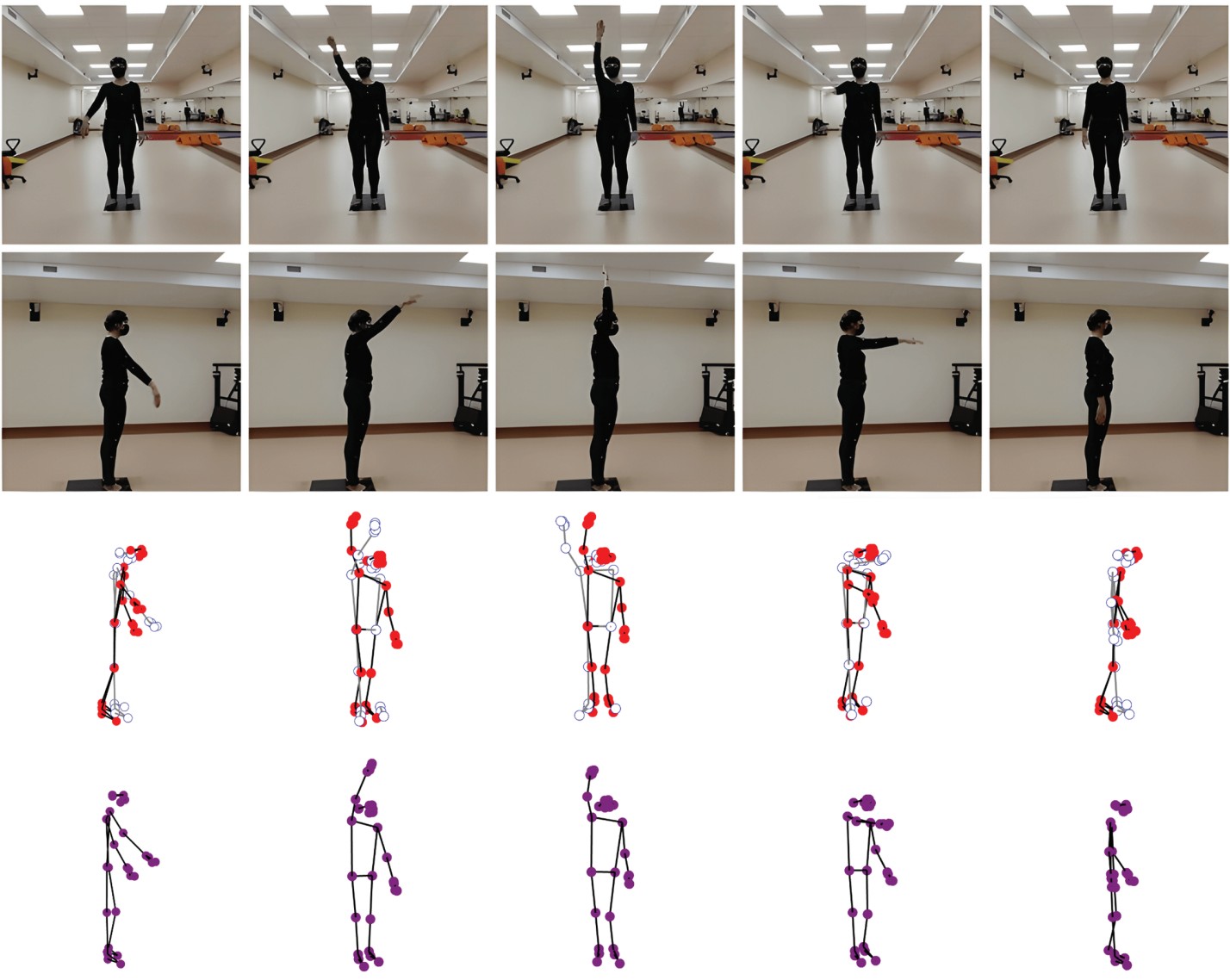

**Fig 3. Illustration of the result of the 3D human skeleton for flexion of the upper arm in the first data set.**

where $N$ is the number of video frames; $x1$, $x2$ – the absolute error of the three-dimensional human skeleton joint depth in the frame between the flow predicted from the front view and the final solution; $\mu$ is the mean of the absolute value of difference of the 3D human skeleton joint depth error over the entire video.

With additional data from the second video stream, the data from the side camera successfully managed to help to adjust the depth coordinates of the human skeleton. Different tests of the same exercises showed that the final results are predicted almost identically and the time variation of the depth coordinates coincides, in contrast to the data from only one video stream.

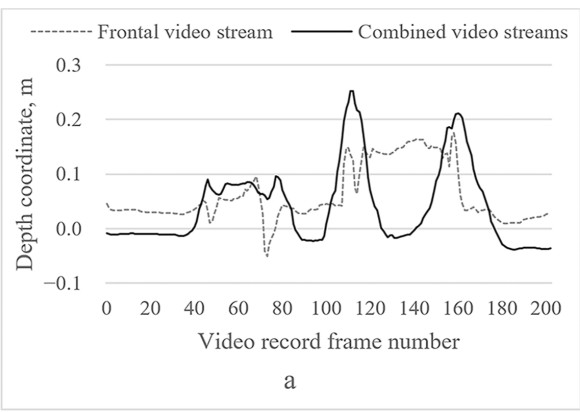 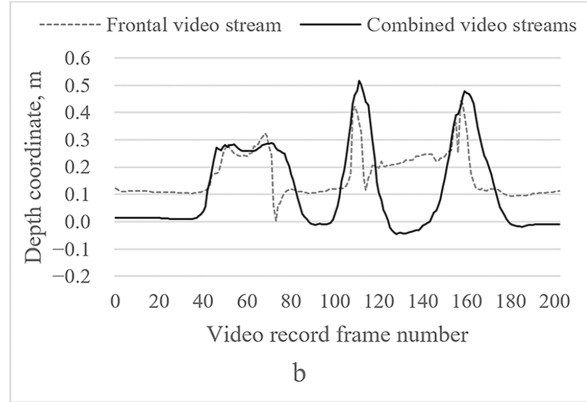

**Fig 4. Illustration of the variation of the depth coordinates in the first test of the upper arm flexion of the first data set.** a – right elbow joint; b – right wrist joint.

**Table 4. Mean and standard deviation of predicted joint depth error for the first data set.**

| Exercise | Right elbow joint | | Right wrist joint | |
|---|---|---|---|---|
| | **Mean, m** | **SD, m** | **Mean, m** | **SD, m** |
| Standing flexion of the upper arm no. 1 | 0.0611 | 0.0446 | 0.1128 | 0.0756 |
| Standing flexion of the upper arm no. 2 | 0.0708 | 0.0391 | 0.1225 | 0.0648 |
| Standing flexion of the upper arm no. 3 | 0.0855 | 0.0652 | 0.1344 | 0.0964 |
| Standing flexion of the upper arm no. 4 | 0.0770 | 0.0483 | 0.1275 | 0.0699 |
| Standing Arm Retraction No. 1 | 0.0803 | 0.0463 | 0.1127 | 0.0745 |
| Standing Arm Retraction No. 2 | 0.0752 | 0.0446 | 0.1214 | 0.0647 |
| Standing Arm Raise No. 2 | 0.0614 | 0.0482 | 0.0884 | 0.0876 |
| Standing Arm Raise No. 3 | 0.0582 | 0.0494 | 0.0857 | 0.0857 |

## Results using the 'Second' dataset

Fig 5 illustrates the predicted three-dimensional human skeleton during an arm flexion exercise. The first two rows display five essential frames from the front and side video streams, respectively. The third and fourth rows present the corresponding 3D skeleton reconstructions: red filled circles represent the front stream projection, aligned empty circles show the side-stream projection, and pink coordinates denote the final output of our proposed solution.

These 3D results are displayed from three angles: side, front, and diagonal. The skeleton projected from the frontal video stream exhibits depth errors, specifically an upper body tilt forward and a lower body tilt backward. Additionally, the skeleton projected from the side stream lacks discernible joints on the left side of the human body. While our final solution successfully straightens the 3D human skeleton and corrects right-hand movements, a depth error persists in the left hand.

Fig 6 presents the variation in depth coordinates for the right elbow and wrist joints during an arm flexion exercise, under different clothing conditions (light vs. dark). These joints, central to right-handed exercise, were selected for their significant depth variations. The results indicate that, while the dual-camera system effectively corrects depth errors, clothing differences introduce minor variations in joint detection accuracy.

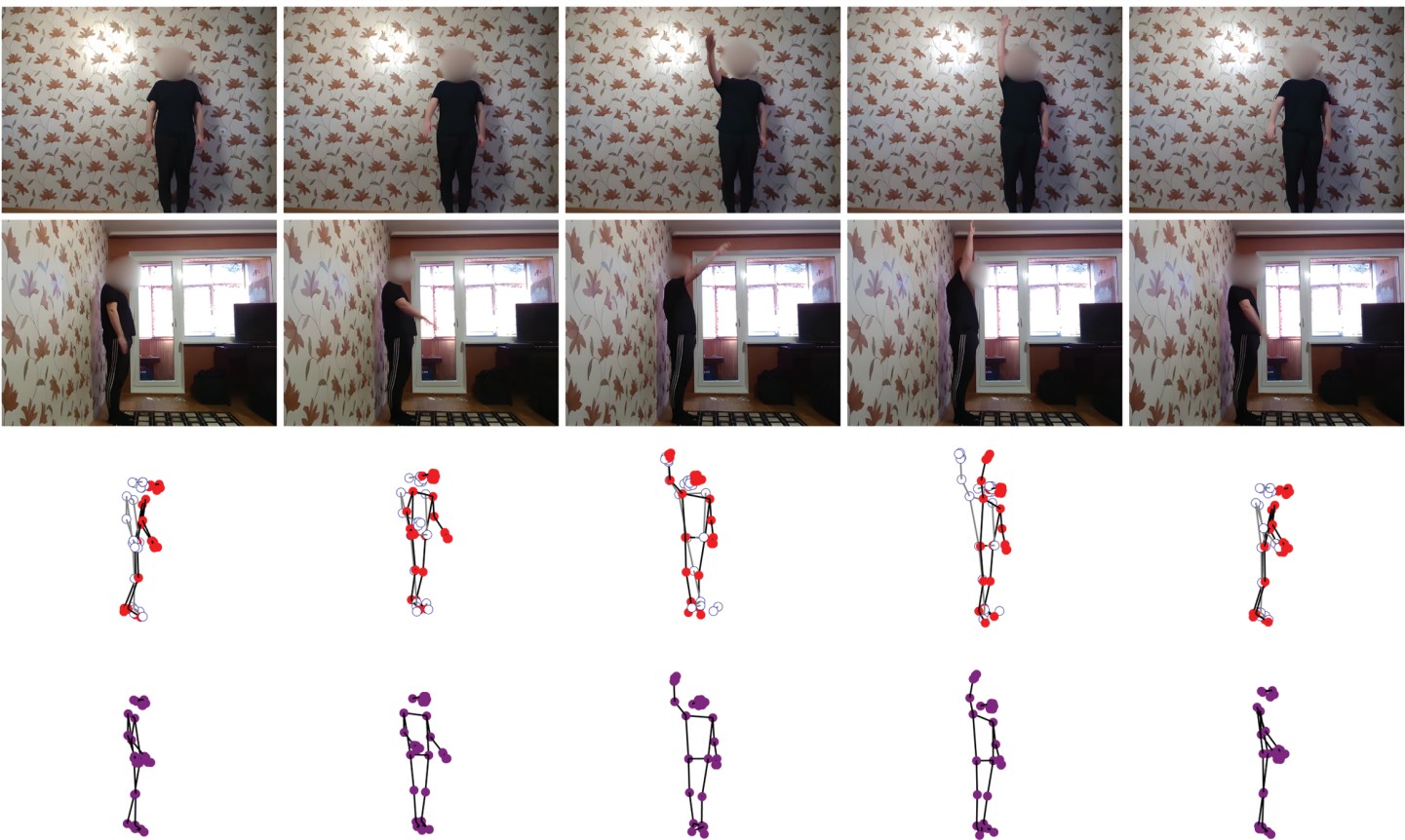

**Fig 5. Illustration of the 3D human skeleton result of flexion of the upper arm in the second data set.**

The initial 70 frames represent system synchronization, ensuring view alignment before the exercise begins. As the arm is raised, depth errors increase, particularly with dark clothing, likely due to reduced contrast and feature detection challenges. Despite this, the final corrected depth trajectory remains stable, with its smoothness and consistency demonstrating the algorithm's robustness in mitigating depth estimation errors and maintaining tracking accuracy across varying visual conditions.

The largest predicted depth error occurs when both hands are raised when a person is dressed in dark clothing. The trajectory of the change in the depth of the right and left elbow and wrist joints remains similar, only the distance differs. The motion trajectory of the final corrected 3D human skeleton remains similar to the test of this exercise in light clothing.

The means and standard deviations of the predicted joint depth error for the second data set are shown in Table 5 and are given in meters.

As with the experiments on the first dataset, additional data from the second image stream successfully corrected the depth coordinates of the 3D human skeleton. Different clothing did not have a significant effect on the change in joint depth trajectories over time, and similar results were obtained compared to the first data set. Both visual 3D space representations and depth comparison plots show that a second image stream perpendicular to the first image stream from the front has a positive effect on the final 3D skeleton prediction. Experiments performed on selected datasets improved depth prediction by an average of 0.1 meters.

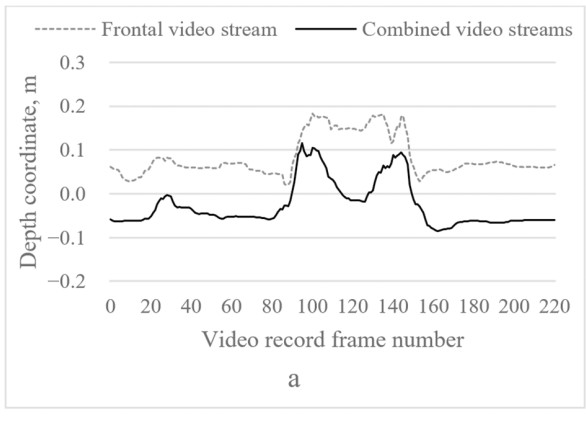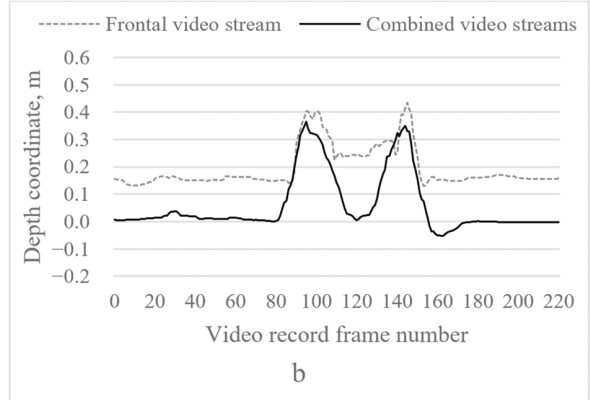

**Fig 6. Illustration of the change of depth coordinates of the second data set of upper arm flexion in light clothing.** a – right elbow joint; b – right wrist joint.

**Table 5. Mean and standard deviation of predicted joint depth error for the second data set.**

| Exercise | Right elbow joint | | Right wrist joint | | Left elbow joint | | Left wrist joint | |
|---|---|---|---|---|---|---|---|---|
| | Mean, m | SD, m | Mean, m | SD, m | Mean, m | SD, m | Mean, m | SD, m |
| Standing arm flexion, light clothing | 0.1104 | 0.0292 | 0.1384 | 0.0475 | - | - | - | - |
| Standing arm flexion, dark clothing | 0.1349 | 0.0518 | 0.1454 | 0.0621 | - | - | - | - |
| Standing arm adduction, light clothing | 0.1325 | 0.0375 | 0.1803 | 0.0496 | - | - | - | - |
| Arm retraction while standing, dark clothing | 0.1637 | 0.0489 | 0.2027 | 0.0564 | - | - | - | - |
| Arm adduction while standing, light clothing | 0.1089 | 0.0406 | 0.0927 | 0.0552 | - | - | - | - |
| Arm adduction while standing, dark clothing | 0.1051 | 0.0596 | 0.1251 | 0.0830 | - | - | - | - |
| Standing biceps curl, light clothing | 0.1041 | 0.0365 | 0.1275 | 0.0553 | 0.0850 | 0.0293 | 0.0750 | 0.0295 |
| Standing biceps curl, dark clothing | 0.1275 | 0.0495 | 0.1477 | 0.0651 | 0.0917 | 0.0376 | 0.0817 | 0.0385 |

### Direct performance comparison

Table 6 presents a comparison of our proposed method with state-of-the-art 3D pose estimation models, evaluated on the Human3.6M dataset. Although recent approaches such as SlowFastFormer achieve the lowest MPJPE, our method provides a competitive performance (41.2 mm) while introducing a dual-camera setup with Kalman filtering and depth error correction.

Table 7 summarizes the mean and standard deviation of the depth error for the right elbow and wrist in two upper limb exercise datasets (Tables 4 and 5), comparing our proposed

**Table 6. Comparison of Pose Estimation Models using Human3.6M dataset.**

| Model | MPJPE (mm) |
|---|---|
| PoseFormer (2021) | 51.4 |
| HybrIK (2021) | 41.6 |
| LiftFormer (2022) | 40.2 |
| GAST-Net (2022) | 39.8 |
| Sosa-self (2023) | 43.4 |
| SlowFastFormer (2024) | 38.9 |
| Hourglass Tokenizer (2024) | 39 |
| MotionAGFormer (2024) | 38.4 |
| **Our Method (2025)** | 41.2 |

**Table 7. Comparison benchmark models, aggregated across all exercises.**

| Model | First Dataset | | Second Dataset | |
|---|---|---|---|---|
| | Elbow (Mean ± SD) | Wrist (Mean ± SD) | Elbow (Mean ± SD) | Wrist (Mean ± SD) |
| PoseFormer | 0.120 ± 0.030 | 0.188 ± 0.046 | 0.176 ± 0.040 | 0.186 ± 0.042 |
| HybrIK | 0.104 ± 0.109 | 0.142 ± 0.045 | 0.129 ± 0.049 | 0.144 ± 0.071 |
| SlowFastFormer | 0.090 ± 0.015 | 0.164 ± 0.035 | 0.166 ± 0.025 | 0.129 ± 0.031 |
| Proposed Method | 0.071 ± 0.048 | 0.113 ± 0.077 | 0.106 ± 0.039 | 0.112 ± 0.047 |

model with three state-of-the-art baselines (PoseFormer, HybrIK, and SlowFastFormer), all of which were benchmarked using only a single frontal video stream. In particular, our proposed model consistently achieves low mean errors (e.g., $0.071 \pm 0.048$ m elbow error in the first dataset) while maintaining competitive or better results relative to the benchmark methods in both datasets.

## Performance analysis and ablation study

The real-time applicability of the proposed approach was evaluated using the same system setup, which was used to collect the data sets. The average computation time of full pipeline was 18.4 ms on desktop PC. Table 8 summarizes an ablation and performance study in which the main algorithmic blocks were replaced by a lightweight alternative. When the Kalman filter was turned off and the missing joints were filled with last held value; this reduces the latency from 18.4 ms to 15.5 ms, but increased the mean elbow error from 7.5 mm to 8.9 mm and the wrist error from 11.3 mm to 13.1 mm. Substituting the linear regression model with raw depths yielded a reduced inference time (17.1 ms) and a small loss of accuracy. Replacement of spatial and temporal synchronization with a nearest-neighbor timestamp match delivered the fastest inference (13.1 ms). However, the estimation error increased to 11.4 mm for the elbow and 15.6 mm for the wrist.

## Statistical analysis

To evaluate the effectiveness of our dual-camera approach in reducing depth estimation errors, we conducted a paired t-test comparing depth errors under different conditions:

- Single-camera vs. dual-camera setup;
- Unobstructed vs. partially obstructed joints;
- Frontal vs. non-frontal views.

Table 9 summarizes the mean depth errors for each condition, along with the corresponding p-values and results of statistical significance.

The results show a significant reduction in depth estimation errors when using the dual-camera setup compared to a single-camera approach ($p < 0.01$). The mean depth error

**Table 8. Performance and ablation metrics.**

| Component Evaluated | Inference Time | Elbow (Mean ± SD) | Wrist (Mean ± SD) |
|---|---|---|---|
| Without Kalman filter | 15.5 ms | 0.089 ± 0.058 | 0.133 ± 0.082 m |
| No linear regression (raw depths) | 17.1 ms | 0.08 ± 0.054 | 0.12 ± 0.078 m |
| Spatial and temporal sync disabled | 13.1 ms | 0.114 ± 0.051 | 0.156 ± 0.076 m |
| Full pipeline (proposed) | 18.4 ms | 0.075 ± 0.042 | 0.113 ± 0.066 m |

**Table 9. Statistical comparison of mean depth error across conditions.**

| Comparison | Depth Error (m) | p-value |
|---|---|---|
| *Single-Camera vs. Dual-Camera* | | |
| Condition 1: Single-Camera | 0.110 | $4.42 \times 10^{-4}$ |
| Condition 2: Dual-Camera | 0.078 | |
| *Unobstructed vs. Partially Obstructed Joints* | | |
| Condition 1: Unobstructed | 0.090 | $1.25 \times 10^{-4}$ |
| Condition 2: Partially Obstructed | 0.129 | |
| *Frontal vs. Non-Frontal Views* | | |
| Condition 1: Frontal View | 0.098 | $3.05 \times 10^{-6}$ |
| Condition 2: Non-Frontal View | 0.118 | |

decreased from 0.110 m (single-camera) to 0.078 m (dual-camera), demonstrating the advantage of leveraging a secondary camera for improved depth prediction.

Similarly, occlusion conditions had a significant effect on tracking accuracy ($p < 0.01$). The mean depth error increased from 0.090 m (unobstructed) to 0.129 m (partially obstructed joints), indicating that occlusions introduce noticeable tracking errors. However, even under occluded conditions, the dual-camera approach helped reduce the impact of missing joint information. The upper limb occlusion conditions were simulated using a dual-camera setup, where the frontal camera captured the full body view of the subject, while the side camera, positioned at 90°, provided depth information. The joints became partially obstructed due to self-occlusion (e.g., arms crossing the torso) or camera perspective limitations, affecting visibility in one of the views. The system mitigated these occlusions by synchronizing both views and interpolating missing joint positions.

Lastly, the results confirm that the camera perspective plays a key role in depth accuracy. Depth errors were significantly lower when the view was frontal (0.098 m) compared to non-frontal perspectives (0.118 m, $p < 0.01$). This suggests that while our spatial synchronization method mitigates some perspective-related errors, certain joint positions may still be more accurately estimated from a frontal viewpoint.

These findings highlight the robustness of our method in reducing depth errors and maintaining tracking accuracy under different conditions, supporting its application in rehabilitation and motion analysis scenarios.

## Discussion

Markerless motion tracking is a highly promising and in-demand field due to easy progress monitoring and real-time feedback. Continuous monitoring allows monitoring patient progress in real-time, adjusting new protocols as needed. Visual feedback from skeletal tracking systems can motivate patients to remain engaged in their programs and enable remote monitoring and rehabilitation, allowing subjects to perform exercises at home. An essential problem with most existing markerless human posture estimation systems and their application in rehabilitation facilities is their level of accuracy and reliability [21], finding that many factors influence the results, beginning with deviation parameters (joint coordinates, joint angles, joint depth error) and ending with varying levels of accuracy, making it challenging to compare results unambiguously with those obtained by other researchers. However, we also achieved a result similar to other approaches (see Table 6).

Table 7 shows that our proposed model outperforms the latest approaches in terms of mean joint depth error, particularly for the elbow and wrist joints. However, such significant improvement is achieved only in our collected dataset. The key factor contributing to

this improvement is our dual-camera setup with temporal and spatial synchronization, which significantly reduces depth ambiguity and errors related to upper limb occlusion, as confirmed in previous research on multiview skeletal tracking [22]. The Kalman-filtering-based depth correction further refines the predictions, mitigating inconsistencies in the estimation of single-camera poses. Interestingly, in the Human3.6M dataset, our model achieves results comparable to other methods (MPJPE: 41.2 mm), despite its enhanced performance in our dataset. This similarity can be attributed to the fact that Human3.6M primarily consists of single-view recordings, limiting the advantages of our multi-camera integration.

Our current benchmarking was performed using a single frontal camera stream. While our proposed model demonstrates strong performance in this context, it is important to acknowledge that the benchmarked baselines, like PoseFormer and HybrIK, could also potentially achieve improved 3D pose estimation by incorporating a multi-camera setup.

In this study, we used linear regression to relate the vertical joint position to the depth error because of its simplicity and real-time efficiency. Despite promising results, human joint trajectories, even for simple rehabilitation exercises, are inherently non-linear [40]. Muscle actuation, joint limits, and multi-segment kinematics produce velocity and acceleration profiles that vary smoothly but not linearly over time. For future work, and given the need for real-time performance in rehabilitation, simple non-linear methods such as Gaussian process regression [41] or kernel ridge regression [42] could be investigated. Alternatively, small multilayer perceptrons or temporal convolutional networks can learn non-linear mappings directly from data [43].

A key advantage of our timeframe-synchronisation module is that it is tuned with Pareto optimisation, which treats alignment accuracy ($E_{sync}$) and computational cost ($C$) as competing objectives rather than forcing a single weighted sum. As a result, the optimizer returns a front of non-dominated solutions that expose the inherent trade-off: every millisecond shaved from processing latency pushes synchronization error up, and vice versa. This explicit trade-off view also guards against hidden overfitting: if future deployments involve slower hardware or denser video streams, the same Pareto front can be recomputed, and a new operating point selected without retraining the full model. Achieved 18 ms latency at 0.078 m mean depth error proved to be the most practical, but the choice of any point ultimately depends on the application or clinical priorities.

Despite promising results, certain factors limit its accuracy. Different subject profiles, such as children, frail elderly people, or orthotic wearers, can distort limb proportions and cause pose errors. Secondly, fluctuating illumination or cluttered backgrounds may suppress key-point confidence and could bias depth estimation. Our configuration assumes the ideal 90° angle between cameras, precise orientation and height consistency (±1cm), any actual deviation or incomplete perspective distortion may lead to systematic errors. Furthermore, our synchronization method mainly compensates for rotations around the Y-axis and corrects potential out-of-plane rotations with insufficient compensation. Unmodeled image blurs can also affect the accuracy of joint detection and depth estimate. Therefore, future work could consider using explicit calibration methods with established methods (such as Zhang's method [44] or fiducial marker-based calibration like AprilTags [45]) to mitigate distortion and alignment errors. However, our simplified approach is practical and emphasizes the ease of reproducibility and deployment, especially for clinical or home rehabilitation.

Furthermore, performing intrinsic camera calibration under simplified assumptions can introduce inaccuracies due to inherent lens distortion effects (specifically, radial and tangential distortions) [46]. Thus, inferring focal length from the manufacturer's stated horizontal field of view and image width contributes to additional error. Small discrepancies between the nominal field of view and the actual in-scene focal length can result in scale errors of up

to ±5% [47]. Those scale errors are likely directly propagating into reconstructed 3D joint coordinates, causing consistent underestimation or overestimation of depth across the entire skeleton. Thus, any unknown focal length mismatch remains uncorrected, which may be particularly problematic when subjects perform exercises at varying distances from the cameras.

By integrating pose refinement algorithms such as SmoothNet [48], FLK [49], or HiMoReNet [50], into our proposed dual-camera pipeline, we could offer refined smoothness and accuracy of the skeleton trajectories. Such post-processing modules usually are designed as a 'plug–and–play' as input takes a sequence of 2D/3D joint coordinates and output smoothed poses by modeling long-range temporal relations per joint. For example, SmoothNet can be applied to each joint channel individually, which is particularly appealing in rehabilitation context, although additional refinements would be needed for real-time applications. On the other hand, FLK combines an adaptive Kalman filter with a biomechanical constraint adjustment and a low-pass filter, and offers a real-time join refinement solution for a desktop PC. HiMoReNet uses a hierarchical architecture that group joints and captures long-range temporal context, and then refines each group's motion while leveraging global body interactions. Thus, using the HiMoReNet module in our proposed pipeline could enforce biomechanically plausible trajectories, which is crucial, if the full system would be applied for montion quantification or anomaly detection.

Although this study evaluated the system using upper limb rehabilitation movements (e.g., arm adduction, flexion, and brachial motions), the framework is inherently scalable to full-body or lower-limb activities. By adjusting camera placement, calibration parameters, and pose estimation targets, it can be adapted for tracking gait, posture, or multi-joint exercises, enabling broader rehabilitation applications. To increase the clinical relevance of the system, future works should include validation against standardized rehabilitation outcome measures such as the Motor Assessment Scale (MAS) and the Disabilities of the Arm, Shoulder, and Hand (DASH) questionnaire. These tools are well suited to evaluate upper limb motor function and the practical usability of motion tracking systems in clinical rehabilitation. Pilot studies involving patient cohorts could further assess the integration of this approach into therapeutic protocols and provide feedback on the effectiveness and usability of the system in real-world clinical environments.

## Conclusions

Our proposed three-dimensional human skeleton tracking methodology has demonstrated the ability to refine upper limb depth coordinates for greater accuracy by fusing a 90° side view with the frontal stream through automatic spatio-temporal alignment, Kalman smoothing and lightweight depth-error regression. By merging the two image streams, the system reduces depth-coordinate errors in wrist and elbow prediction with automatic spatial and timeframe synchronizations. The fusion of predicted human skeleton models allows building a human skeleton model even when one camera does not see all parts of the human body, and we can refine the depth coordinates using the error prediction from the linear regression model. The prediction of depth coordinates from a single image has errors that exhibit a regular deviation along the vertical axis. However, using an additional 90° rotated video camera directed at the person, we can compensate for the inaccuracies of depth prediction and reduce errors by 0.4 m.

Although the dual-view layout relies solely on commodity RGB cameras and lightweight calibration, providing a cost-effective path to continuous DASH/MAS scoring, remote telerehabilitation, and straightforward extensibility to full-body or gait analysis via simple camera repositioning, reliable claims about its accuracy still require rigorous implementation and

benchmarking against existing rehabilitation-oriented systems. In particular, these solutions must be validated in real-world clinical scenarios with established assessment instruments before they can be trusted for routine motion assessment workflows in physiotherapy and rehabilitation.

## Author contributions

**Conceptualization:** Kristina Daunoravičienė, Artūras Serackis, Rytis Maskeliūnas.

**Data curation:** Vytautas Abromavičius, Kristina Daunoravičienė, Jurgita Žižienė.

**Formal analysis:** Rytis Maskeliūnas.

**Funding acquisition:** Vytautas Abromavičius.

**Methodology:** Vytautas Abromavičius, Ervinas Gisleris, Artūras Serackis, Rytis Maskeliūnas.

**Project administration:** Rytis Maskeliūnas.

**Resources:** Vytautas Abromavičius, Kristina Daunoravičienė, Jurgita Žižienė.

**Software:** Vytautas Abromavičius, Ervinas Gisleris.

**Supervision:** Kristina Daunoravičienė, Artūras Serackis, Rytis Maskeliūnas.

**Validation:** Vytautas Abromavičius, Kristina Daunoravičienė, Rytis Maskeliūnas.

**Visualization:** Ervinas Gisleris.

**Writing – original draft:** Vytautas Abromavičius, Jurgita Žižienė, Rytis Maskeliūnas.

**Writing – review & editing:** Vytautas Abromavičius, Rytis Maskeliūnas.

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
