## [Decision Letter · Decision Letter 0]

2 Jun 2025

PONE-D-25-15851Robust Skeletal Motion Tracking Using Temporal and Spatial Synchronization of Two Video StreamsPLOS ONE

Dear Dr. Abromavičius,

Thank you for submitting your manuscript to PLOS ONE. After careful consideration, we feel that it has merit but does not fully meet PLOS ONE’s publication criteria as it currently stands. Therefore, we invite you to submit a revised version of the manuscript that addresses the points raised during the review process.

The manuscript presents a dual-camera skeletal tracking system enhanced by spatial-temporal synchronization, linear regression-based depth correction, and Kalman filtering, aimed at improving depth accuracy for rehabilitation monitoring. Both reviewers find the approach well-structured and relevant, with meaningful innovation in its integration of simple models for real-time, low-cost setups. However, they raise concerns about the limited novelty relative to existing stereo-vision systems, lack of system scalability, insufficient real-time performance metrics, and the oversimplification of depth correction through linear regression. Please include the references, mentioned by the reviewer(s), if any, relevant to the topic.The authors are suggested to address all the concerns carefully. 

We look forward to receiving your revised manuscript.

Kind regards,

Jyotindra Narayan

Academic Editor

PLOS ONE

Journal Requirements:

3. Thank you for stating the following financial disclosure: [This project has received funding from the Research Council of Lithuania (LMTLT), agreement No S-PD-24-29.]. 

4. Thank you for stating the following in the Acknowledgments Section of your manuscript: [This project has received funding from the Research Council of Lithuania (LMTLT), 461

agreement No. S-PD-24-29.]

Reviewers' comments:

Reviewer's Responses to Questions

**Comments to the Author**

1. Is the manuscript technically sound, and do the data support the conclusions?

Reviewer #1: Yes

Reviewer #2: Yes

2. Has the statistical analysis been performed appropriately and rigorously? 

Reviewer #1: Yes

Reviewer #2: Yes

3. Have the authors made all data underlying the findings in their manuscript fully available?

Reviewer #1: No

Reviewer #2: Yes

4. Is the manuscript presented in an intelligible fashion and written in standard English?

Reviewer #1: Yes

Reviewer #2: Yes

5. Review Comments to the Author

Reviewer #1: The paper presents a novel dual-camera skeletal tracking system aimed at improving depth accuracy in human pose estimation for rehabilitation monitoring. Positioned within the context of low-cost, real-time rehabilitation applications, its main contribution is the integration of temporal and spatial synchronization, a linear regression-based depth error correction model, and Kalman filtering to enhance robustness and accuracy.

While multi-camera setups are not new, the focus on dynamic depth error correction via linear regression, combined with Kalman filtering, offers meaningful innovation. The work is well-motivated, addressing a critical barrier to affordable home-based skeletal tracking.

The paper details the synchronization, regression correction, and Kalman filtering with clear mathematical formulations. However, some aspects need clarification: (1) the linear regression model may oversimplify complex 3D depth errors; exploring or justifying more sophisticated models would strengthen the approach; (2) calibration and alignment assumptions are critical but only briefly discussed; deeper elaboration on calibration errors and mitigation would aid reproducibility. The methodology is sound but would benefit from a more thorough discussion of limitations.

Experiments are well-documented, with both quantitative metrics and qualitative visualizations demonstrating superior depth estimation accuracy over baseline methods. Still, improvements are suggested: (1) include more diverse baselines (e.g., recent transformer-based models); (2) add ablation studies to isolate the contributions of each component (e.g., regression vs. Kalman filtering).

A discussion of practical applications and integration with pose refinement algorithms, such as:

- http://dx.doi.org/10.48550/arXiv.2112.13715

- https://doi.org/10.1109/LSP.2023.3295756

- https://doi.org/10.1016/j.sigpro.2024.109598

would further enhance the paper.

Reviewer #2: The manuscript presents a well-organized and timely contribution to the field of skeletal motion tracking, particularly targeting rehabilitation scenarios. The integration of a dual-camera system, along with spatial and temporal synchronization mechanisms such as linear regression-based depth correction and Kalman filtering, reflects a thoughtful approach to addressing known limitations of single-camera systems. The proposed method is validated across custom datasets and benchmarked against established models, showing promising improvements in depth accuracy. The clarity of presentation and the thorough methodology are commendable. The work holds strong potential for applications in telerehabilitation and home-based monitoring. However, there are several areas where the manuscript could be further strengthened:

1. While the paper introduces a dual-camera system, the novelty is somewhat incremental given existing literature on stereo vision and multi-view pose estimation. The manuscript would benefit from a clearer articulation of how the presented system differs fundamentally in architecture, methodology, or application domain from prior approaches.

2. The real-time applicability of the system is emphasized, yet there is no quantitative evidence provided to substantiate this claim. Details on processing time per frame, latency, and computational requirements should be reported, especially considering the use of multiple video streams and filtering algorithms.

3. Discuss on how the system would generalize to broader scenarios—such as diverse subject profiles, variable lighting, or real-world rehabilitation environments.

4. The Kalman filter implementation is well described mathematically, but the choice and tuning of parameters (e.g., noise covariance matrices) are not discussed. Providing this information is essential for reproducibility and for understanding the system’s adaptability across different motion patterns.

5. The Pareto optimization technique used for temporal alignment is innovative but lacks intuitive explanation. A visual diagram or step-by-step example would be beneficial, particularly for readers from non-engineering backgrounds.

6. Although comparisons with state-of-the-art models like PoseFormer and SlowFastFormer are presented, it is not fully clear if these baselines were evaluated under the same dataset and preprocessing conditions. Greater transparency in the benchmarking methodology is recommended.

7. The claim of robustness against occlusion is supported in theory, but the experimental validation is somewhat limited in this regard. Additional test cases with realistic occlusions (e.g., arms crossing the body, partial visibility) would help validate this important aspect.

8. Depth correction is achieved using a linear regression model, which may be effective in structured tasks but could lack flexibility in more dynamic or non-linear motion scenarios. Exploring the potential of non-linear regression techniques could be valuable.

9. Visualizations are helpful but could be refined for accessibility. The color-coded skeletons are difficult to distinguish when printed in grayscale. Including markers or labels could improve figure readability across different formats.

10. Some redundancy is observed in the text, particularly in repeated phrases describing depth distortion patterns. Condensing these sections would improve flow without diminishing clarity.

11. The manuscript frames the work within a rehabilitation context, yet it lacks discussion on how this system could be integrated with clinical protocols or validated using standardized outcome measures. Even a brief exploration of future clinical validation would increase the paper's practical relevance.

12. The scope of the method appears limited to upper-limb motion, but the paper does not address whether this framework can be extended to full-body or lower-limb activities. A short note on system scalability would provide helpful context for broader application.

6. PLOS authors have the option to publish the peer review history of their article (what does this mean?). If published, this will include your full peer review and any attached files.

Reviewer #1: No

Reviewer #2: No

---

## [Author Response · Author response to Decision Letter 1]

18 Jun 2025

Journal Requirements:

3. Thank you for stating the following financial disclosure: [This project has received funding from the Research Council of Lithuania (LMTLT), agreement No S-PD-24-29.].

4. Thank you for stating the following in the Acknowledgments Section of your manuscript: [This project has received funding from the Research Council of Lithuania (LMTLT), 461

agreement No. S-PD-24-29.]

Answer.

Thank you for investing your time and attention in handling our submission. We have carefully considered every point raised in the decision letter and the accompanying reviews, and we have revised the manuscript accordingly: our methodological distinctions, refining the figures for grayscale accessibility, expanding the Results and Discussion to make our contributions to upper-limb rehabilitation unmistakable. We believe these edits fully address the reviewers and your recommendations and improve the clarity and impact of the work.

We have reviewed the requirements, and made small adjustments over all manuscript, especially about the affiliations section. We have included additional information about the funding. Indeed, the funding status matches the statement: ""The funders had no role in study design, data collection and analysis, decision to publish, or preparation of the manuscript."". Also, we have included this information in cover letter. And we would like to request the editor, to include funding information in the official Funding Statement with the manuscript. Moreover, full ethics statement is now included in the Methods section.

Reviewer #1: The paper presents a novel dual-camera skeletal tracking system aimed at improving depth accuracy in human pose estimation for rehabilitation monitoring. Positioned within the context of low-cost, real-time rehabilitation applications, its main contribution is the integration of temporal and spatial synchronization, a linear regression-based depth error correction model, and Kalman filtering to enhance robustness and accuracy.

While multi-camera setups are not new, the focus on dynamic depth error correction via linear regression, combined with Kalman filtering, offers meaningful innovation. The work is well-motivated, addressing a critical barrier to affordable home-based skeletal tracking.

The paper details the synchronization, regression correction, and Kalman filtering with clear mathematical formulations. However, some aspects need clarification:

(1) the linear regression model may oversimplify complex 3D depth errors; exploring or justifying more sophisticated models would strengthen the approach;

Answer. We agree that our linear model may underfit inherently non-linear joint dynamics. We believe, that sophisticated models would require additional research, since the processing time could influence systems capabilities for deployment. We have updated the Discussion to acknowledge this limitation explicitly and to outline lightweight non-linear alternatives—such as Gaussian-process regression, kernel ridge regression, and compact neural models, as promising avenues for future work. However,

(2) calibration and alignment assumptions are critical but only briefly discussed; deeper elaboration on calibration errors and mitigation would aid reproducibility. The methodology is sound but would benefit from a more thorough discussion of limitations.

Answer. We appreciate the reviewer’s insightful comment. Indeed, in our study, we intentionally employed a straightforward calibration and alignment procedure with only minimal manual intervention. We aimed to experiment with a simple dual-camera setup that does not rely on checkboards or other similar solutions. Methods section now clarifies this, and includes a little more detail on the setup, as following:

Both cameras were mounted on tripods at approximately the same height (±1 cm) and oriented to form a nominal 90° angle. We approximate each camera as an ideal pinhole (principal point in the center of the image, no distortion coefficients) using the manufacturer’s horizontal field of view to infer the focal length in pixels.

While acknowledging that this simplicity introduces inherent calibration limitations and should be further discussed. Essentially, it significantly enhances the reproducibility and practical applicability of our approach, particularly in scenarios such as variable user profiles, lightening conditions might impede usability. Thus, we have modified and extended discussion section as following:

Despite promising results, certain factors limit its accuracy. Different subject profiles, such as children, frail elderly people, or orthotic wearers, can distort limb proportions and cause pose errors. Secondly, fluctuating illumination or cluttered backgrounds may suppress key-point confidence and could bias depth estimation. Our configuration assumes the ideal 90 angle between cameras, precise orientation and height consistency ($\pm1$cm), any actual deviation or incomplete perspective distortion may lead to systematic errors. Furthermore, our synchronization method mainly compensates for rotations around the Y-axis and corrects potential out-of-plane rotations with insufficient compensation. Unmodeled image blurs can also affect the accuracy of joint detection and depth estimate. Therefore, future work could consider using explicit calibration methods with established methods (such as Zhang's method or fiducial marker-based calibration like AprilTags to mitigate distortion and alignment errors. However, our simplified approach is practical and emphasizes the ease of reproducibility and deployment, especially for clinical or home rehabilitation.

Furthermore, performing intrinsic camera calibration under simplified assumptions can introduce inaccuracies due to inherent lens distortion effects (specifically radial and tangential distortions). Thus, inferring focal length from the manufacturer’s stated horizontal field of view and image width contributes to additional error. Small discrepancies between the nominal field of view and the actual in‐scene focal length can result in scale errors of up to 5%. Those scale errors are likely directly propagating into reconstructed 3D joint coordinates, causing consistent underestimation or overestimation of depth across the entire skeleton. Thus, any unknown focal length mismatch remains uncorrected, which may be particularly problematic when subjects perform exercises at varying distances from the cameras.

Experiments are well-documented, with both quantitative metrics and qualitative visualizations demonstrating superior depth estimation accuracy over baseline methods. Still, improvements are suggested:

(1) include more diverse baselines (e.g., recent transformer-based models);

Answer. We appreciate this suggestion and have expanded the introduction and results to include two more transformer-based approaches, along to existing transformer-based methods: Hourglass Tokenizer (HoT) and MotionAGFormer. As we have included in the manuscript, those papers report lower MPJPE values than ours. However, we have conducted a direct empirical comparison against a similar transformer-based method SlowFastFormer (2024). As results show our dual-camera pipeline achieves lower elbow and wrist level errors than SlowFastFormer while running in real time.

(2) add ablation studies to isolate the contributions of each component (e.g., regression vs. Kalman filtering).

Answer. We appreciate this suggestion and have added a dedicated ablation study that quantifies the impact of every algorithmic block on both accuracy and inference time. Because several modules are functionally indispensable, the outright removal would break the processing chain. Therefore, we have replaced each one with the simplest viable alternative. For example, Kalman filter was disabled, and any missing joint was filled by the most recent valid estimate; regarding linear regression, the network depth output was directly fed to the fusion stage, and instead of spatial and temporal synchronization we aligned frames solely by the closest time‐stamp.

A discussion of practical applications and integration with pose refinement algorithms, such as:

- http://dx.doi.org/10.48550/arXiv.2112.13715

- https://doi.org/10.1109/LSP.2023.3295756

- https://doi.org/10.1016/j.sigpro.2024.109598

would further enhance the paper.

Answer. We appreciate the reviewer’s suggestion to discuss integration with pose‐refinement methods. In practice, our pipeline can be paired with a lightweight, learned refinement modules. And most of the proposed post-processing modules are suitable for real-time applications like home rehabilitation monitoring, sports analytics or similar. We have updated the discussion section accordingly:

In practical settings, such as clinical rehabilitation monitoring, sports biomechanics, and human-computer interaction, pose estimates may suffer from temporal jitters and occasional large errors (e.g., during occlusions or rare poses). By integrating pose refinement algorithms like SmoothNet, FLK, or HiMoReNet, to our proposed dual-camera pipeline, could offer a refined the smoothness and accuracy of skeleton trajectories. Such post-processing modules usually are designed as a 'plug--and--play' as input takes a sequence of 2D/3D joint coordinates and output smoothed poses by modeling long-range temporal relations per joint. For example, SmoothNet can be applied to each joint channel individually, which is particularly appealing in rehabilitation context, although additional refinements would be needed for real-time applications. On the other hand, FLK combines an adaptive Kalman filter with a biomechanical constraint adjustment and a low-pass filter, and offers a real-time join refinement solution for a desktop PC. HiMoReNet uses a hierarchical architecture that group joints and captures long-range temporal context, and then refines each group’s motion while leveraging global body interactions. Thus, using the HiMoReNet module in our proposed pipeline could enforce biomechanically plausible trajectories, which is crucial, if the full system would be applied for montion quantification or anomaly detection.

Reviewer #2: The manuscript presents a well-organized and timely contribution to the field of skeletal motion tracking, particularly targeting rehabilitation scenarios. The integration of a dual-camera system, along with spatial and temporal synchronization mechanisms such as linear regression-based depth correction and Kalman filtering, reflects a thoughtful approach to addressing known limitations of single-camera systems. The proposed method is validated across custom datasets and benchmarked against established models, showing promising improvements in depth accuracy. The clarity of presentation and the thorough methodology are commendable. The work holds strong potential for applications in telerehabilitation and home-based monitoring. However, there are several areas where the manuscript could be further strengthened:

1. While the paper introduces a dual-camera system, the novelty is somewhat incremental given existing literature on stereo vision and multi-view pose estimation. The manuscript would benefit from a clearer articulation of how the presented system differs fundamentally in architecture, methodology, or application domain from prior approaches.

Answer. Thank you for pointing out the need for clearer differentiation from prior stereo-vision work. In the revised manuscript we now emphasize our contribution in more clear way. There is an updated closing paragraph in the Introduction section. We have also specified, that our main experiments point to upper limb activities (as you pointed out in further comments). Moreover, an expanded Discussion section contrasts these elements with state-of-the-art multi-view methods and existing limitations are discussed and displayed.

2. The real-time applicability of the system is emphasized, yet there is no quantitative evidence provided to substantiate this claim. Details on processing time per frame, latency, and computational requirements should be reported, especially considering the use of multiple video streams and filtering algorithms.

Answer. Thank you for a really important suggestion. We have added a performance analysis that reports inference time on a desktop GPU. The complete pipeline processes each frame well below the 33 ms. Moreover, we have included a small ablation that quantifies the impact of every algorithmic block on both accuracy and inference time. Because several modules are functionally indispensable, the outright removal would break the processing chain. Therefore, we have replaced each one with the simplest viable alternative. For example, Kalman filter was disabled, and any missing joint was filled by the most recent valid estimate; regarding linear regression, the network depth output was directly fed to the fusion stage, and instead of spatial and temporal synchronization we aligned frames solely by the closest time‐stamp.

3. Discuss on how the system would generalize to broader scenarios—such as diverse subject profiles, variable lighting, or real-world rehabilitation environments.

Answer. Thank you for prompting us to elaborate on generalization issues. We have expanded the Discussion with the following text:

Despite promising results, certain factors limit its accuracy. Different subject profiles, such as children, frail elderly people, or orthotic wearers, can distort limb proportions and cause pose errors. Secondly, fluctuating illumination or cluttered backgrounds may suppress key-point confidence and could bias depth estimation. Our configuration assumes the ideal 90 angle between cameras, precise orientation and height consistency ($\pm1$cm), any actual deviation or incomplete perspective distortion may lead to systematic errors. Furthermore, our synchronizatio

---

## [Decision Letter · Decision Letter 1]

10 Jul 2025

Robust skeletal motion tracking using temporal and spatial synchronization of two video streams

PONE-D-25-15851R1

Dear Dr. Abromavičius,

We’re pleased to inform you that your manuscript has been judged scientifically suitable for publication and will be formally accepted for publication once it meets all outstanding technical requirements.

Kind regards,

Jyotindra Narayan

Academic Editor

PLOS ONE

Additional Editor Comments (optional):

As per the feedback and assessmnet from the reviewers and editor, the manuscript is found to be acceptable for the publication. Congratulations to the authors.

Reviewers' comments:

Reviewer's Responses to Questions

**Comments to the Author**

1. If the authors have adequately addressed your comments raised in a previous round of review and you feel that this manuscript is now acceptable for publication, you may indicate that here to bypass the “Comments to the Author” section, enter your conflict of interest statement in the “Confidential to Editor” section, and submit your "Accept" recommendation.

Reviewer #1: All comments have been addressed

Reviewer #2: All comments have been addressed

2. Is the manuscript technically sound, and do the data support the conclusions?

Reviewer #1: Yes

Reviewer #2: Yes

3. Has the statistical analysis been performed appropriately and rigorously? 

Reviewer #1: Yes

Reviewer #2: Yes

4. Have the authors made all data underlying the findings in their manuscript fully available?

Reviewer #1: (No Response)

Reviewer #2: Yes

5. Is the manuscript presented in an intelligible fashion and written in standard English?

Reviewer #1: Yes

Reviewer #2: Yes

6. Review Comments to the Author

Reviewer #1: All comments have been addressed. Methodology is clear, and discussion has been enriched. I suggest the acceptance of the paper.

Reviewer #2: The authors have thoroughly addressed all prior reviewer and editorial comments in their revised manuscript. The responses are comprehensive, and the updated version provides clear methodological clarifications, additional experimental evidence (including ablation studies and performance metrics), improved figures for accessibility, and an expanded discussion on clinical relevance and limitations. Overall, I find that the authors have made substantial improvements, and this version meets the journal’s scientific and presentation standards. I recommend acceptance of the manuscript in its current form.

7. PLOS authors have the option to publish the peer review history of their article (what does this mean?). If published, this will include your full peer review and any attached files.

Reviewer #1: No

Reviewer #2: **Yes: **Subhash Pratap

---

## [Editor Report · Acceptance letter]

PONE-D-25-15851R1

PLOS ONE

Dear Dr. Abromavičius,

I'm pleased to inform you that your manuscript has been deemed suitable for publication in PLOS ONE. Congratulations! Your manuscript is now being handed over to our production team.

Kind regards,

on behalf of

Dr. Jyotindra Narayan

Academic Editor

PLOS ONE